# Conformational plasticity enables functional switching in diatom light-harvesting complexes
Theofani-Iosifina Sousani[1,4], Boutheina Zender [1,4], Sayan Maity [2,3], Ulrich Kleinekathöfer [2] & Vangelis Daskalakis [1] ✉

Biomolecules exhibit a fundamental correlation between structure and function, which can be modulated by environmental factors. Deciphering this relationship remains a central and long-standing challenge for many protein families. In this study, we investigate such a correlation in the light-harvesting complexes (LHCs) of diatoms; unicellular, photosynthetic organisms that thrive in marine ecosystems. Using µs-long molecular dynamics simulations and machine learning, we reveal that all experimentally resolved LHC configurations correspond to a few distinct interconverting states linked to an intrinsic conformational transition that might affect the balance between light-harvesting and photoprotective modes; a property that can be tuned or engineered. Thus, to the best of our knowledge, we provide an original view on the plethora of experimentally resolved structures. Our model strongly correlates with experimental findings on the effect of the photoprotective protein LHCX1 and the xanthophyll cycle on the FCP acclimation states.

Photosynthesis is a vital natural process through which solar energy is captured and transformed into chemical energy, defining the basis of life on Earth[1]. Organisms capable of photosynthesis are generally grouped into green and red evolutionary lineages[2]. Despite their diversity, green plants and algae share common structural features in their photosynthetic systems. Light energy, in the form of photons, is initially absorbed by pigments like chlorophylls and carotenoids found in light-harvesting complexes (LHCs) – protein-pigment assemblies located within the thylakoid membrane of chloroplasts[1]. Varying environmental factors such as changes in light intensity or spectrum, often influenced by daily light cycles, can affect this process. To manage these fluctuations and avoid damage from excess light, green plants and algae use (down) regulatory strategies that adjust light absorption. One key protective response is non-photochemical quenching (NPQ), a mechanism by which surplus excitation energy is safely dissipated as heat to prevent photodamage[3]. Diatoms in the red lineage of photosynthetic organisms are highly important as primary producers in marine ecosystems, supplying organic substrates and oxygen, by rapidly adjusting light harvesting and photochemistry to cell productivity. Therefore, they are considered an ideal model organism for studying photosynthetic regulation[4]. Diatoms express fucoxanthin (Fx) and chlorophyll-a/c (Chl) binding proteins (FCPs) as LHCs. FCPs are responsible for the

exceptional light-harvesting capabilities of diatoms in the blue-green region, which is available underwater. However, they also have a robust intrinsic photoprotective mechanism that enables them to adapt to fluctuating light at the ocean surface[5,6]. The details of this mechanism at all-atom resolution are still obscure[6].

Regulation of photosynthesis is a promising area of research for improving crops and increasing biomass or boosting the atmospheric $CO_2$ assimilation to ease the strangle against global warming[7–10]. The fundamental research on the downregulatory mechanisms of photosynthesis could lay the groundwork for artificial photosynthesis[11], and could also provide crucial knowledge towards the increased production of biofuels and nutrients. FCPs come in monomers or various oligomeric states such as dimers, trimers, tetramers, or even pentamers[5,12–18]. The plethora of structural insights into FCPs calls for a comprehensive investigation to answer the following question: What is the connection between FCP structure and function for a remarkable adaptability? To answer this question, we implemented a comprehensive analytical framework which integrates extensive conformational sampling at the classical Molecular Dynamics (MD) level, exciton coupling calculations between pigments, statistical tools like Markov State Modeling (MSM), data from experimentally resolved structures and machine learning (ML). This integrated computational approach allows for a quantitative assessment of how FCP conformation

[1]Department of Chemical Engineering, University of Patras, Patras, Greece. [2]School of Science, Constructor University Bremen, Bremen, Germany. [3]Department of Physics and Astronomy and Thomas Young Centre, University College London, London, UK. [4]These authors contributed equally: Theofani-Iosifina Sousani, Boutheina Zender. ✉e-mail: vdaskalakis@upatras.gr

changes in response to external stimuli which is likely associated with different acclimation states.

## Results

### The configurational space of light harvesting antennas

We have complemented previous MD simulations on the FCP complex from *P. tricornutum*[14,19] with dynamics of FCP complexes from *C. gracilis* (Sm1 complex in monomeric and tetrameric forms and m2 complex in monomeric forms)[13]. The setup and run of the MD simulations of *C. gracilis* followed exactly the protocols in our previous studies of *P. tricornutum*[19,20]. All FCPs were sampled at two different pH states (neutral and acidic (pH ~5.5) lumenal environments)[19,20] and different multimerization states. For details refer to the Methods section. Our simulations included the complete dimeric, trimeric, or tetrameric FCP complexes within a model thylakoid membrane. However, only for the analysis, we focused on the behavior of individual monomers, being part of oligomers or not. All these together, provided a total of 80 μs of monomer-level dynamics for FCPs from the two species. This ensures that the effects of pH and oligomerization are included in the results. Consistency was ensured in the analysis by truncating all structures to a common core of 111 residues (444 atoms in the mainchain – backbone) that is shared among both diatoms, identified by a multi-sequence alignment method (MUSTANG)[21]. Only residues aa 30–51, 54–116, 132–157 were considered for the FCP from *P. tricornutum*[14]. For *C. gracilis* residues aa 58–79, 83–145, 155–180 were considered for the Sm1 in monomeric and tetrameric forms and aa 58–79, 83–145, 158–183 for the m2 monomer[13]. Interestingly, this minimum core of 111 residues matches a continues sequence in the dimeric FCP fold (H1/ H2) found in the diatom *C. meneghiniana* (aa 64–174)[22]. All residue numbers reported hereafter refer to the numbering in the sequence of H1/ H2.

Cryo-EM (cryogenic electron microscopy) and X-ray crystallographic data were also collected from the protein data bank (pdb). Structures from diverse FCPs in various oligomeric states are available from different species (*P. tricornutum*: 6A2W; *C. gracilis*: 6J3Y, 6J3Z, 6J40, 6JLU, 6L4U, 7VD5, 7VD6, 8WCK, 8WCL; *T. pseudonana*: 8IWH, 8J0D, 8JP3; *C. meneghiniana*: 8J5K, 8J7Z, 8W4O, 8W4P; *C. roscoffensis*: 9KQB). Dimers, tetramers, or trimers were disassembled into 118 different monomeric units before the analysis. Thus, two primary datasets will be analyzed in the following, i.e., set1: MD frames and set2: experimental structures.

A Markov state model (MSM), a statistical analysis tool, was constructed from the MD trajectories, which can predict the long-time scale behavior of a biomolecular system based on short MD simulations[23]. In order to describe the main conformational changes of the FCP complex over data set 1, one can rely on the complete array of backbone torsional angles. However, this huge array contains a lot of noise, i.e., thermal fluctuations and fast motions. Therefore, the time-lagged independent component analysis (tICA) was used to reduce the dimensionality of this array, a method that extracts the slowest motions from the simulations. These motions, captured by two vectors (IC-1 and IC-2), represent the most biologically relevant rearrangements of the protein-pigment complex. IC-1 and 2 consist of linear transformations of the high-dimensionality space to a subspace that maximally preserves the kinetic content. In our case, IC-1 and 2 represent linear combinations of torsional angles on the FCP backbones. For further details refer to the Methods. FES was projected along these vectors of the FCP common core to identify the most stable conformations. This is shown in Fig. 1A, alongside the positions of the four identified kinetically distinct states 1–4, or ensembles of conformations. State-2 is shared between the FCPs of *P. tricornutum* and *C. gracilis*, states-1 and- 3 are assigned to *P. tricornutum* whereas state-4 to *C. gracilis*. Four conformations, as ensemble averages per state, termed macrostates, that are associated with states 1-4 were also predicted by MSM (C1 to C4). The superimposed conformations (macrostates) of the common core for C1-4 are shown in Fig. 1B. Therein, the FCP scaffold clearly shows a progressive expansion, or swelling mainly at the lumenal side, from conformation C1 and C4 to C2 and finally to C3. The mechanism and functional significance of this expansion are not readily available. However, the transitions should be

related to pH differences at the lumen FCP side or oligomerization states, given that the MD trajectories sampled FCP complexes at different pH values and oligomeric states (see Methods for FCP model setup). Given also that the combination of MD-MSM provides insight into kinetically distinct conformations of a biomolecular system over the long-time scale[23], C1-4 should indicate key conformational transitions of the FCP scaffold not explicitly sampled by the classical MD.

The tICA components or vectors cannot be easily understood in terms of physical parameters that can describe conformational changes of the FCP protein scaffold. We thus have employed an alternative to the tICA dimensionality reduction approach. The configurational space of the MD set1 was reduced into two dimensions: (a) the average (avg.) angle $\bar{\theta}$ between helices in the common FCP core and (b) the average of the tilts of the same helices with respect to the Z axis. The Z axis is defined as the normal on the thylakoid membrane (see Fig. 1B for definitions). The avg. angle $\bar{\theta}$ describes the relative orientations of key transmembrane helices within the FCP complex, providing a measure of their concerted bending or splaying. The avg. tilt quantifies how these helices are oriented relative to the membrane plane, thereby reflecting potential conformational shifts that could modulate pigment-pigment interactions relevant for light harvesting and photoprotection. Moreover, the avg. angle $\bar{\theta}$ and avg. tilt might also capture transitions between partial helicity or random coil structures, especially for the lumenal side of the FCP complex (vector $v_2$) as these transitions would shift the avg. values. Together, these parameters summarize protein conformational changes, like scaffold expansion (or swelling), that can influence pigment interactions. The probability distribution P(x,y) of the MD data set1 over the two reduced dimensions is converted into a free energy landscape by -ln(P(x,y)) (energy in units of kT) (Fig. 1C). This representation also highlights the most stable states of the protein, which correspond to the minima on the surface. The landscape in Fig. 1C shows roughly the same conformational space as in Fig. 1A but arranged in a rotated orientation.

To validate our results, we determined the relationship between the predicted C1-4 conformations and the experimentally FCP resolved structures across species. We repeated the dimensionality reduction procedure for data set2 and the results are shown in Fig. 1D. The position of the MSM-predicted conformations or macrostates C1-4 are shown for reference on both plots (Fig. 1C, D). Remarkably, the theoretical predictions quantitatively match the minima also in the distribution of the experimentally resolved FCP structures from the protein databank (set2). The C1-4 conformations do not correspond to minima sampled by MD (Fig. 1C), as they are long timescale "projections" based on the MD-MSM combination that, however, they remarkably fit the experimental data, even for different species of diatoms. In Fig. 1E we show the distribution of each experimentally resolved structure colored by species over the avg. angle and avg. tilt dimensions. As can be observed, there is no dependency of the C1 to C4 conformations on the diatom species. We have to note that the distribution of Fig. 1E is also independent of the resolution of the experimentally resolved structures.

For further analysis, we have used machine learning approaches, namely algorithms that can detect patterns in complex data like in an elaborate fitting process, without needing an explicit equation. Supervised models rely on a well-established set of input features (like the avg. angle and avg. tilt), whereas unsupervised models do not need the definition of specific features to identify patterns. By training these models on our simulation results (input features), we can uncover relationships (output) between protein structure and pigment interactions that would be difficult to capture with simple analytical approaches. For details refer to the Methods section. The experimentally resolved set2 structures (Fig. 1E) were classified (grouped) into four clusters identified by color (Fig. 1F) and based on their structural diversity. This was done by unsupervised classification which was performed by a simple k-means clustering over the avg. angle and avg. tilt features. K-means clustering is a way of grouping similar protein conformations together based on their structural features. The algorithm looks for patterns in the data and assigns each conformation to one of several clusters, such that conformations within the same cluster are more similar to

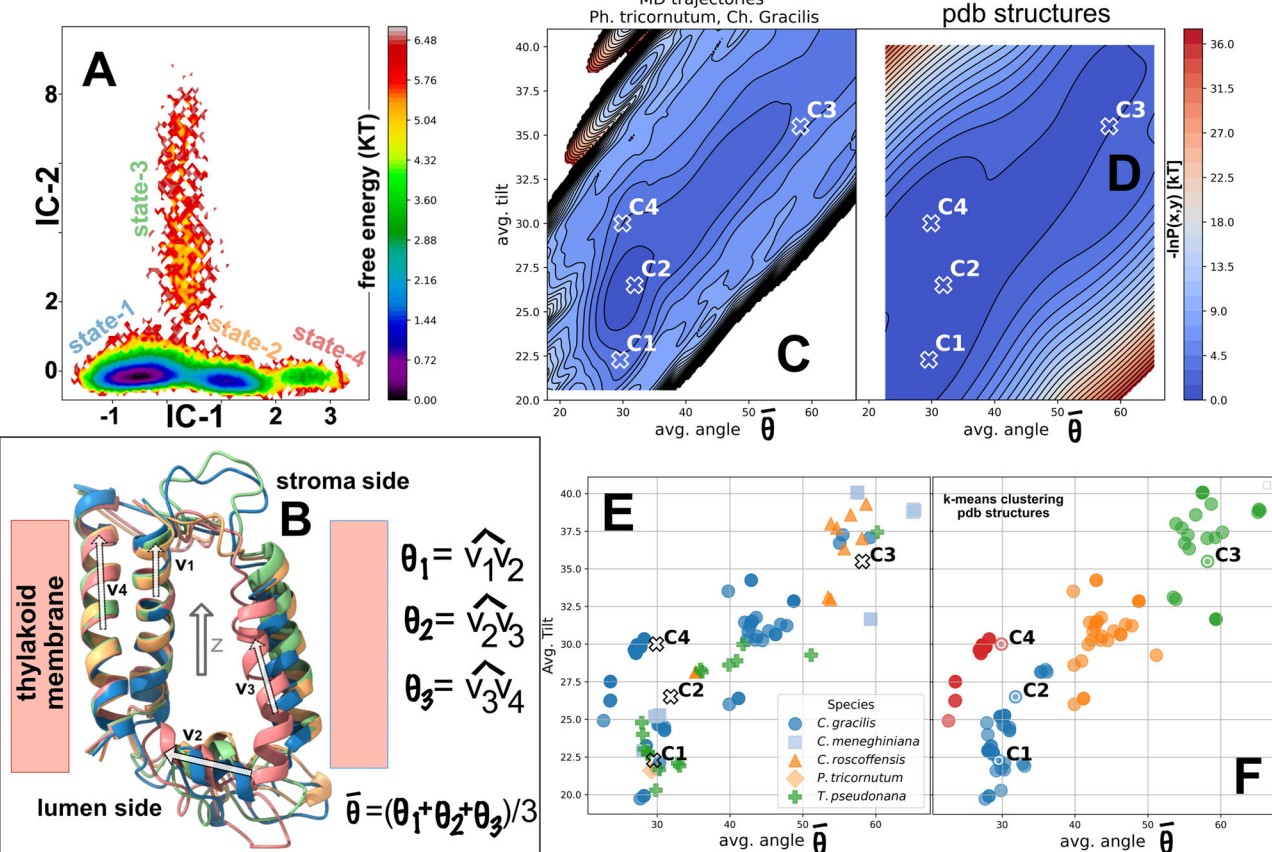

**Fig. 1 | The configurational space of the diatom light harvesting complexes.**
**A** Free Energy of the protein scaffold spanned over the tICA components IC-1 and IC-2. The four main energy states (1 to 4) are highlighted with colored labels denoting their position. The energy values are given in units of kT with k being the Boltzmann constant and T the temperature. **B** The FCP monomer core in the different C1 − 4 conformations associated with states 1–4 color coded as in (**A**). The avg. angle $\bar{\theta}$ is defined as the average of the angles $\theta_1$, $\theta_2$ and $\theta_3$ defined by the vectors: $v_1$ aa 64-85, $v_2$ 109–116, $v_3$ 117–131 and $v_4$ for aa 149–174. The avg. tilt is calculated

as the average of the angles formed by each vector $v_1$ to $v_4$ with respect to the membrane normal Z. The residue numbering refers to the sequence of the H (H1/H2) dimer of *C. meneghiniana*. (**C**) Energy distributions of the FCP scaffold conformations spanned over the avg. angle and tilt vectors for the MD data set1 and (**D**) the experimentally resolved structural data set2. **E** The distribution of experimentally resolved structures per species. **F** Clustering of the experimentally resolved structural data set2 into four color-coded clusters. In (**C–F**), the position of C1-4 conformations is given for reference.

each other than to those in other clusters. In practice, this means that instead of analyzing millions of individual simulation frames, we can summarize the dynamics into a handful of representative structural 'states,' which correspond to the cluster centers. C1, C2 belong to the same cluster (blue), whereas C4 to the red cluster and C3 to the green cluster. The classification is not always obvious by just placing an unknown structure alongside the data of Fig. 1F. However, a more robust supervised machine learning model called Random Forest (RF) was trained on the experimentally resolved structures along with the four MSM-predicted conformations (C1-4) and verified the original classification (Fig. 1F). RF approach is like asking many independent non-linear fitting algorithms ('decision trees') the same question (e.g., the value of a y(x) dependent variable) and then averaging their answers for the output. Each tree looks at the data, or fits the data in a slightly different way, and together they provide a reliable consensus. This helps reduce the chance of over-fitting to noise and highlights which structural features most strongly affect FCP conformations. The C1 to C4 MSM-predicted conformations, are thus ultimately classified into three main clusters (red, blue and green). The trained RF ML model can thus be employed to characterize accurately any future FCP structure providing also percentages of "blue", "red", "orange" or "green" character.

## Structure and function correlation

Could the distinct C1-4 conformations be associated with a robust transition of the FCP scaffold between an efficient light-harvesting and a downregulated

photosynthetic state? In the absence of strong experimental evidence on the actual NPQ site (specific pigments) within FCPs[24], we can rely only on theoretical predictions and an experimental study[19,25,26]. We focus on a pigment pair in the FCP structure of *P. tricornutum* (Chl-a 409 and carotenoid Fx-301) that has been proposed computationally as the only pigment pair to exert pH-dependent fluctuating excitonic coupling values between 7 and 40 cm$^{-1}$ and experimentally to be involved in an important triplet Chl decay channel[19,25,26]. Under NPQ conditions, energy transfer between Chl-a and carotenoids (Cars) like Fx or Dtx could enable a quick dissipation of the excess absorbed energy as heat, through a short-lived excited state of Car, in an analogy to higher plants where the major LHC of Photosystem II binds the Chl-a 612/ Lutein pigment pair and energy can be dissipated by energy transfer from Chl-a to Lutein[27,28]. The excitonic coupling value between Chl-a (m) and Car (n) is a measure of the efficiency of this energy transfer, with the rate of transfer given by $k_{mn} \approx |V_{mn}|^2$ [19], where $V_{mn}$ is the excitonic coupling between m and n. The $V_{mn}$ values for the pigments of this pair have been calculated in an earlier study on the FCP from *P. tricornutum* from a 860 snapshot trajectory[19] and herein for *C. gracilis* Sm1 monomer (861 snapshots randomly extracted). Low $V_{mn}$ values (<10 cm$^{-1}$) indicate rather weak Chl-Car interactions and higher values (>10 cm$^{-1}$) are characteristic for efficient energy transfer, that might relate with a component of the NPQ dissipating mechanism[19]. In fact, aggregation of FCPs significantly enhanced excitation-energy quenching by changes also in Chl-Car interactions that affected energy transfer dynamics, as probed by absolute

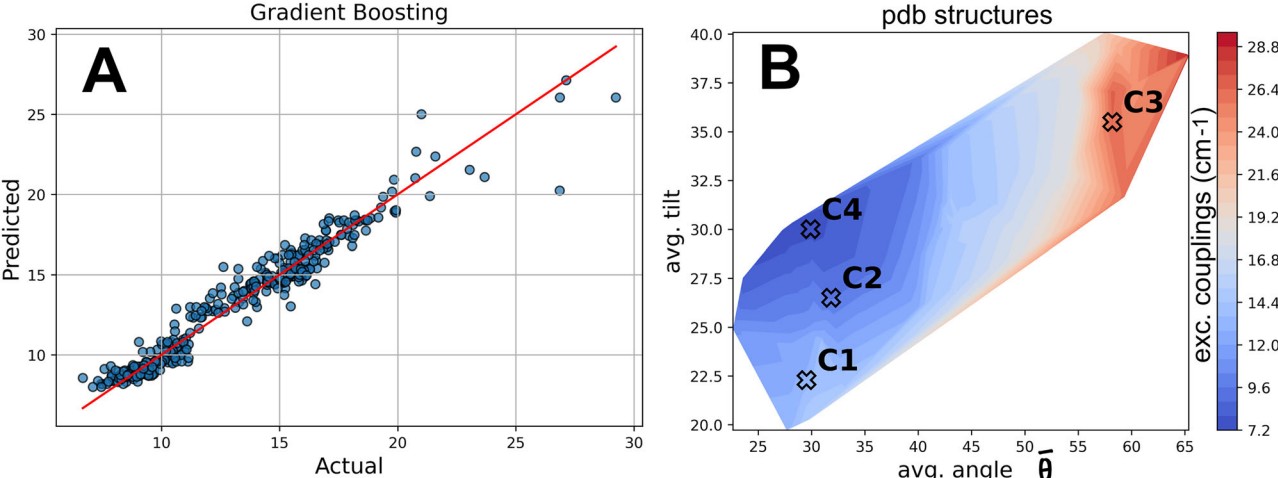

**Fig. 2 | Correlation between structure and function. A** Evaluation of the regression by gradient boosting for the calculated excitonic couplings for the Chl-a 409/ Fx-301 pigment pair (cm⁻¹) in the MD data. **B** Predicted distribution of excitonic couplings

spanned over the avg. angle and avg. tilt dimensions for the experimentally resolved structural data set.

fluorescence spectroscopy and fluorescence decay-associated spectra[29]. The authors of the latter study also suggest that Chls may interact with nearby fucoxanthins, resulting in excitation quenching.

A gradient boost regression (GBR) algorithm was trained on the features (avg. angle and avg. tilt) of the combined data set (1721 snapshots) and the associated excitonic couplings for Chl-a 409/ Fx-301 as output target. Gradient boosting regression also uses decision trees, but in this case each tree is built step by step, learning from the mistakes of the previous one. This gradual refinement allows the model to capture subtle relationships in the data, making it powerful for predicting how small structural changes influence inter-pigment coupling.

Feature-target relationships were visualized by plotting the predicted versus the actual values for 1721 snapshots of both species[19] as shown in Fig. 2A. This plot indicates a very strong correlation between geometric characteristics of the helices (angles, tilts) – the input features – and the excitonic couplings, at least for the FCP models in the two species of the MD data set1[19,20]. The trained GBR model was employed to predict "equivalent" coupling values based on the two structural features (avg. angles and avg. tilts) for the whole experimental data set2 in this study (Fig. 2B) as well as the C1-4 conformations. A clear trend is identified with an increase of excitonic couplings in the transition from C4 (7.65 cm⁻¹) to C2 (9.48 cm⁻¹), to C1 (13.07 cm⁻¹) and to C3 (26.41 cm⁻¹). The coupling values roughly follow the progression of the FCP scaffold expansion or swelling as depicted in Fig. 1B (C1-C4 to C2 and to C3). We find that certain conformations slightly enhance the interaction between Chl-a 409 and Fx-301 pigments, by an increase in Chl-Fx excitonic coupling values. While this alone does not prove that quenching occurs for C3, it points to a structural mechanism that likely enhances energy dissipation under photoprotective conditions. It is possible that other inter-pigment excitonic couplings within FCPs are also important for the transition of the complex between different light acclimation states (see also discussion in the section "*proof of concept*"). The assignment of excitonic coupling values for other pigment pairs within FCPs, or the calculation of excited state lifetimes is, however, beyond the scope of the current study and impossible at present as the exact NPQ sites have not been identified yet for all the different FCP variants and thus no definite experimental proof is available. Thus, our current model might not be able to unambiguously assign all experimentally resolved FCP structures to an exact acclimation state. However, it is remarkable that a general trend is obvious with our model capturing quantitative general patterns for the FCP conformation, that can be qualitatively correlated with excitonic couplings for a specific pair of pigments.

## An intrinsic flexibility of the FCP scaffold

For further analysis, the MSM-predicted C1-4 conformations mapped onto the sequence of the minimum core of the H dimer in *C. meneghiniana* (aa 64–174) were fed into ProteinMPNN; a deep learning protein design algorithm[30]. The algorithm predicts new protein sequences that fold into the desired 3D protein structure (C1-4). Only the aa 109–131 (sequence (seq): VAAINAIPALGWAQIIFAIGAVD) were allowed to be designed (mutated) so that the end sequence could fold to the associated C1-4 conformation with contracted or expanded (swollen) protein scaffolds. The resulting sequences of the highest score were folded by AlphaFold[31] as shown in Fig. 3A and referred to as AF1 (seq: PRGVLAFPPALQALLAPVAALLH), AF2 (seq: PARHAILDPWAVLPAVLALLVVL), AF3 (seq: PLALLALT-PEQRALLAAILGALL) and AF4 (seq: LAALPYIDPALWERVAAA-LAELE) for C1, C2, C3 and C4 based predictions, respectively. Despite the failure in reproducing the exact expansion - contraction trend of C1-4 in Fig. 1B, the predictions show clearly contracted and expanded scaffolds with the same variability. RF/ GBR models predict that AF4 belongs to the orange cluster (Fig. 3B) with an excitonic coupling at 16.01 cm⁻¹, just in-between C1-C2-C4 and C3 (Fig. 1F). "Equivalent" excitonic couplings are also predicted for AF1 at 12.48 cm⁻¹, AF3 at 9.14 cm⁻¹, and AF2 at 17.77 cm⁻¹. Despite the supposed expansion of the AF1 scaffold (shown in blue in Fig. 3A), expansion is only occurring on the stromal side. In contrast, the lumenal side appears to be contracting. This AF1 conformation is classified within the same cluster as AF3. The ability of ProteinMPNN to design distinct sequences compatible with the different C1-4 conformations suggests that these conformations are not only physically plausible but also intrinsically supported by the FCP protein scaffold. Given that the ProteinMPNN and AlphaFold are trained on a wide range of experimentally validated structures, this result points to an inherent structural plasticity of the FCP fold likely enabling conformational tuning through sequence variation, pH, or oligomerization state. This result highlights the intrinsic conformational versatility of the FCP fold that stably adopts its various experimentally observed conformations.

## Proof of concept

We must note that the NPQ mechanism in diatoms is highly dependent on the xanthophyll cycle and the presence of the LHCX family of photoprotective proteins[32,33]. Our MD models do not consider these effects thus far, yet we could reproduce all different FCP conformations found in the protein data bank. This could imply that the xanthophyll cycle and LHCX proteins fine tune the populations of these FCP conformations in vivo. To prove our case, we have run additional simulations for the FCP from *P. tricornutum* in complex with the photoprotective LHCX1 protein and

**Fig. 3 | Sequence dependent configuration.**
**A** Equivalent to C1-4 conformations, the AF1-4 conformations predicted by ProteinMPNN and AlphaFold, are shown along with their position (**B**) on the two dimensions (avg. angle, avg. tilt). The coloring is identical to the one used in Fig. 1B.

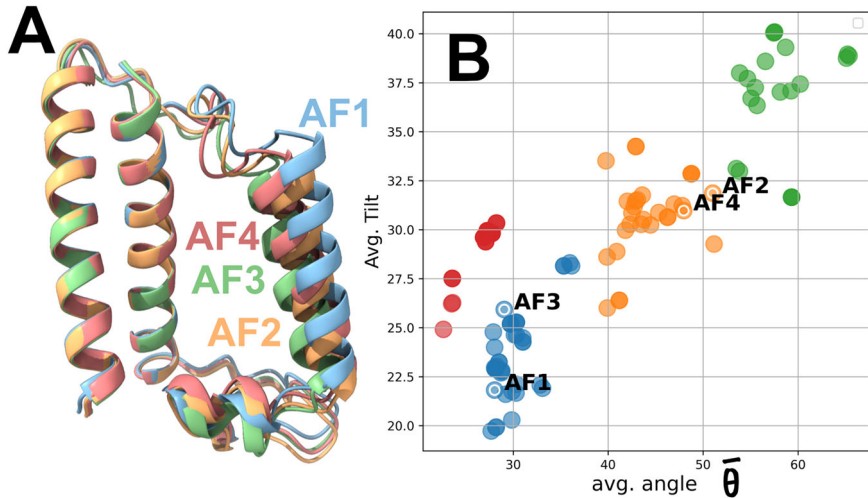

we also exchanged diadinoxanthin (Ddx) to diatoxanthin (Dtx) at low lumenal pH to take into consideration the xanthophyll cycle[34,35]. We have modeled an LHCX1 structure without pigments. Modeling a fully pigmented LHCX1 would require a separate study, since the pigment composition and precise pigment binding locations, If any, are not yet well established in the literature. Such an effort would involve building multiple structural models with different pigment arrangements, supported by biochemical assays and spectroscopic data from isolated LHCX1. We plan to address this pigmented model in a future work. The Ddx-Dtx exchange was only considered in the models where FCP interacts with LHCX1 at low pH. For FCP-LHCX1 model setup (Fig. 4A) please refer to the Methods section. In Fig. 4B we show the distribution of FCP conformations for isolated FCPs at both low pH and neutral pH (Ddx present) and for FCP-LHCX1 at neutral pH (Ddx present) and at low pH (Dtx present). A comparison of Figs. 4B and 2B shows that the LHCX1 protein and the xanthophyll cycle activates the strong interaction between Chl-a 409 and Fx-301 by shifting the FCP population to expanded (swollen) FCP scaffolds and increased excitonic coupling values for the Chl-a 409/Fx-301 pigment pair (cm$^{-1}$) in the diatoms *P. tricornutum* and *C. gracilis*. Furthermore, we observe that upon the FCP-LHCX1 interaction at low pH, the Dtx assumes various orientations (red structures in Fig. 4C) that are associated with the C3 conformation as shown in Fig. 4C. The Dtx orientation in the presence of LHCX1 is significantly more variable as also shown from the distribution of RMSD values as violin plots in Fig. 4D with respect to the FCP Cα atoms used as reference. Ddx orientation shows less variance between isolated FCPs at low and neutral pH (orange structures in Fig. 4C, D). Here, Dtx is modeled at the interface between LHCX1 and FCP (Fig. 4A, C) based on the original location of Ddx in the experimentally resolved crystal structure of the FCP from *P. tricornutum*[14]. Therefore, Dtx interactions with adjacent pigments can be tuned by the interaction with LHCX1. Chl-a 405 is found adjacent to Ddx/ Dtx thus LHCX1 can tune also the energy transfer dynamics between Chl-a 405 and Dtx. Moreover, the RMSF values for the carbon atoms of Ddx/Dtx indicate significant mobility in the transition from low pH (Ddx) to low pH (LHCX1+Dtx) indicated by the Δ(LHCX1, Dtx) plot at low pH in the inset of Fig. 4D, along with the superposition of the main Ddx/Dtx conformations along the dynamics sampled. Recent experimental evidence suggests that Dtx displays significant conformational changes for high-light treated FCPa upon aggregation[24]. Experimental studies in the literature[6,34–37] also indicate that LHCX1 and Dtx are necessary components for the induction of NPQ in diatoms. In summary, our results show that these components also fine tune the dynamics of the FCP scaffold and pigment interactions therein. In the case of a pigmented LHCX1, the different Dtx orientations (Fig. 4C, D) could activate energy dissipating pathways between Chls bound in LHCX1 and Dtx.

## Discussion

This study addresses the fundamental biochemical relationship between protein structure and function in photosynthesis. Specifically, it seeks to answer the question: how are photosynthesis and light absorption regulated in terms of the conformations of LHCs? Using microsecond-scale all-atom molecular dynamics, Markov state modeling, and machine learning (ML), we demonstrate that all experimentally determined fucoxanthin and chlorophyll a/c-binding protein (FCP) structures, the LHCs in diatoms, correspond to only a few interconverting conformational states. Overall, our analysis grouped the protein conformations into four main states, each separated by slow transitions. These states represent distinct structural arrangements that the FCP can adopt during its natural dynamics, or the transition between light-harvesting and photoprotective states. Our simulations provide a molecular-level complementary to experimental observations of the FCP plasticity. In line with the experimental literature[38], we have identified that the light harvesting antennas in diatoms (FCP scaffold) are quite flexible. While time-resolved spectroscopy and cryo-EM have revealed static structures and ensemble energy transfer dynamics, our approach captures the conformational fluctuations and inter-pigment exciton couplings that could underlie such processes. FCPs undergo expansion (or swelling) and contraction in both Molecular Modeling for just two species (*P. tricornutum* and *C. gracilis*) and for all experimentally resolved structures across species.

Experimentally resolved structures of FCPs consist mainly of multimers (Fig.1D–F). In fact, although more balanced, also considerably more data from multimers are collected compared to monomers for the MD-based dynamics shown in Fig. 1C; for the same sampling of 2 μs, we collect 2 μs dynamics for the monomer, but 8 μs monomer-equivalent dynamics for a tetramer. Thus, the Free Energy Surfaces shown in Fig. 1 are more biased towards multimers. However, the FES shown in Fig. 4B even though it refers strictly to monomer only dynamics, yet it shows the same picture as the ensembles biased more towards the multimers. Furthermore, in our previous study[19], we have shown that FCP monomers and multimers can share the same states. Cumulatively, these results point to a flexible FCP scaffold that can be fine-tuned by pH, LHCX1-Dtx and multimerization states, with this tuning to be leaving its signature on the experimentally resolved structures. Such FCP flexibility, or conformational changes, appear to be associated with inter-pigment excitonic couplings, at least within FCPs from *P. tricornutum* and *C. gracilis* and correlate with experimental works on the crucial effect of the photoprotective LHCX1 protein and the xanthophyll cycle. Our study aligns well with a recent experimental work on FCP aggregates that show fluorescence quenching[24]. In this latter study, under NPQ conditions, Dtx exerts considerably stronger conformational changes compared to Ddx in line with Fig. 4C, D herein. We go beyond this observation to identify the mechanism behind these changes as being the

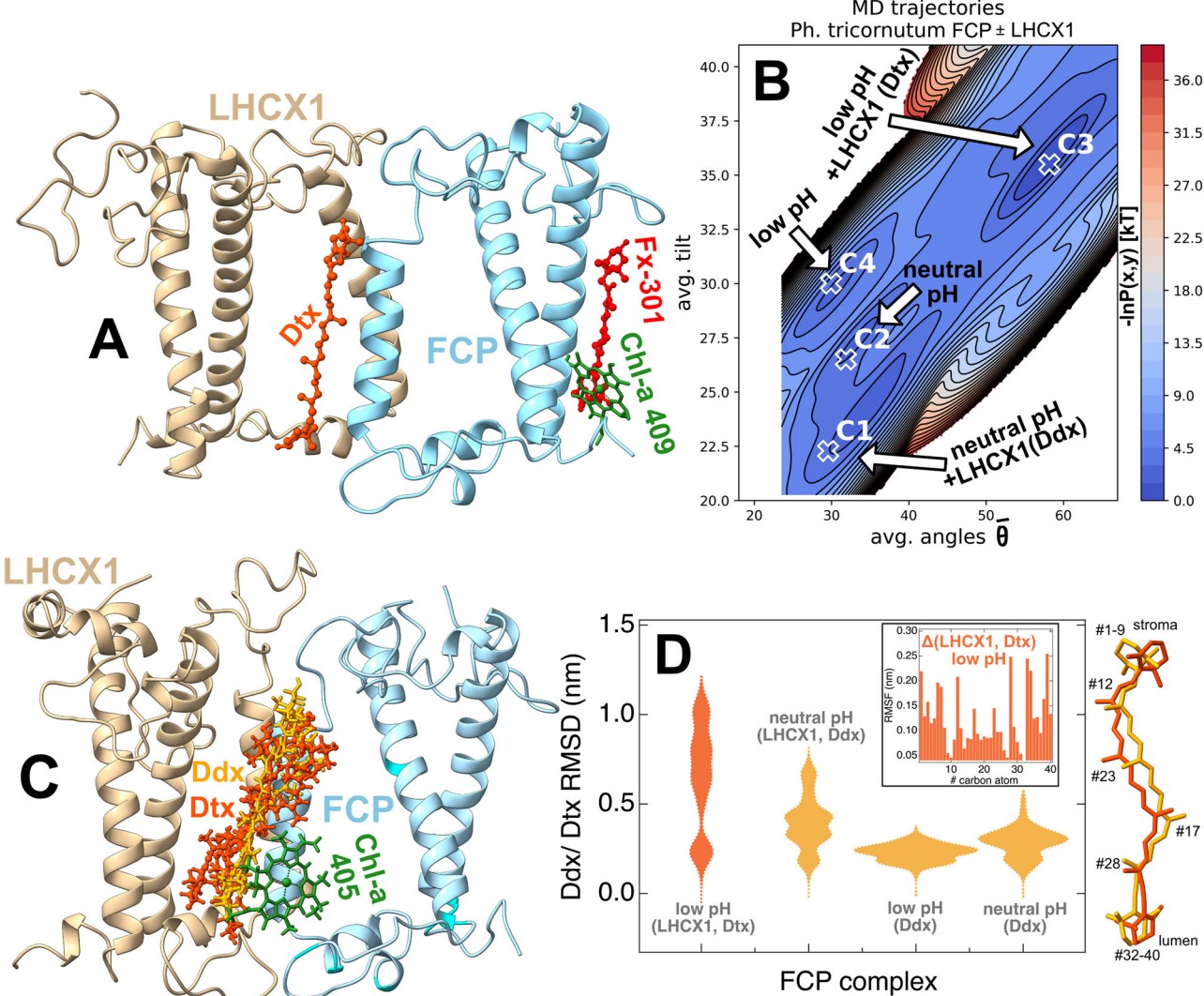

**Fig. 4 | The effect of LHCX1 and the xanthophyll cycle. A** The structure of the FCP-LHCX1 complex. Key pigments are shown for reference (diadinoxanthin (Ddx) or diatoxanthin (Dtx), Chlorophyll (Chl) 409 and Fucoxanthin (Fx) 301). **B** Energy distributions of the FCP scaffold conformations spanned over the avg. angle and tilt dimensions for models of data set1 and FCP-LHCX1 (Dtx - diatoxanthin) at low pH and FCP-LHCX1 (Ddx - diadinoxanthin) at neutral pH complexes, **C** Orientations of the Ddx (orange) and Dtx (red) carotenoids over the dynamics in the absence of LHCX1 interaction (Ddx) and the presence of LHCX1 interaction (Dtx). Chl-a 405

in the vicinity of Ddx/Dtx is also shown for reference, **D** Distributions of the root mean square deviations (RMSD) of the carbon atoms of Ddx (orange) or Dtx (red) as violin plots for the different models. The inset shows the Root Mean Square Fluctuations (RMSF) of the Dtx/ Ddx carbon atoms for Δ(LHCX1, Dtx). The atom numbering is shown for selected regions on two representative structures: Ddx (orange)-Dtx (red); the ring close to the stroma #1-9, the ring close to the lumen #32–40 and branching carbons.

FCP-LHCX1 interaction or possible FCP-FCP aggregation with the same interface as the FCP-LHCX1 complex[25] (see also the methods section). We propose that experimentally resolved structures capture intermediate or transition states of FCP protein dynamics as a fingerprint of the FCP acclimation state. This is the first demonstration that the structural heterogeneity or plasticity of FCPs[24] reflects an intrinsic, functionally tunable well described conformational landscape.

The simulations suggest that small shifts in the relative position (angle, tilt) of FCP helices can change how strongly chlorophylls and fucoxanthins interact. This provides a possible structural link between protein plasticity and the balance between light harvesting and photoprotection. We emphasize that our results highlight a possible conformational change that is associated with increased coupling between chlorophylls and fucoxanthins. We note that experimental evidence has also suggested that Fx carotenoids contribute to quenching[24,29]. We observe a modest increase in Chl–Fx coupling (from 7.65 to 26.41 cm⁻¹), which however translates into a $\left(\frac{26.41}{7.65}\right)^2 \approx 12$ fold increase of the exciton

transfer rate and also suggests a structural predisposition toward enhanced interaction between these pigments. However, stronger coupling alone does not ensure NPQ. Thus, our simulations should be interpreted as identifying a structural mechanism that could contribute to quenching, rather than defining it. Future combined computational-experimental work will be needed to assess whether the observed conformational states also shift the energetic landscape of Chl and Fx/Dtx in a way that enables efficient energy dissipation. Our predictions could be experimentally tested using site-directed mutagenesis that can bias helix packing by introducing either Cys pairs for cross-linking so that the FCP complex can assume expanded (swollen) scaffold, or by replacing bulky residues to Ala to favor a contracted scaffold. Measurements like Transient Absorption (TA) and Two-Dimensional Electronic spectra (2DES) can identify Chl-Fx exciton energy transfers therein. Furthermore, by collecting FCP/LHC sequences from public genomes sequence alignments to build phylogenies and associate such sequences with AlphaFold structural predictions. This would provide helix-packing metrics and

applying phylogenetically aware statistics/machine learning one can associate sequence signatures with expanded or contracted scaffolds and diatom phenotypes (efficient light harvesters or with robust photoprotection). For example, species found under sea ice (polar) are adapted to even lower, blue-shifted light, while open-water species are better equipped to cope with light fluctuations and rapid changes, demonstrating diverse phenotypes among diatoms.

In the context of this study, we have developed algorithms that can be used to classify FCP structures resolved experimentally into different clusters (groups), or to evaluate those predicted computationally from different diatom species. Our algorithms can be refined as more MD or crystallographic data become available. They can also be used in terms of workflow to identify key conformations in other protein families. It should be noted that the acclimation state for FCP structures has been established based on the calculated excitonic coupling of a Chlorophyll-fucoxanthin pigment pair within FCPs from *P. tricornutum* and *C. gracilis*. This approach may be limited in its application to FCP complexes from other diatom species, or even on *P. tricornutum* and *C. gracilis* as other pigments might also be involved, for example pigments bound to LHCX1. Nevertheless, we find a strong correlation between the computational and experimental results regarding the effect of the photoprotective LHCX1 protein and the xanthophyll cycle on the state of the FCP in *P. tricornutum*, which positively validates our approach. Nevertheless, our findings highlight the need for further investigation.

## Methods

### Setup of FCP models

The crystal structure of the fucoxanthin chlorophyll a/c -binding (FCP) protein from the diatom *C. gracilis* (pdb: 7vd5)[13] was used as the initial coordinates to build the monomer models (Sm1 and m2). To construct the S-tetramer, we used the monomer structure of Sm1 and superimposed it with the tetramer chains Sm1, Sm2, Sm3, Sm4 from the Nagao et al. structure[13]. Although chain Sm2 in their structure exhibits slight differences compared to the other three, a previous study[15] refers to the S-tetramer as an homotetramer. All polypeptide chains were described using the Amber ff14SB force field[39]. Amber-compatible parameters for fucoxanthin and DD6 were taken from different studies[39–41], while chlorophyll a and c parameters were adopted from two separate studies[42,43]. The missing phytyl tails of chlorophyll a were modeled using Schrödinger Maestro to restore the complete molecular structure. Protonation states for the Sm1 monomer were assigned as follows: all lumen-facing Asp and Glu residues were protonated to reflect pH 5.5[19]. In contrast, at pH 7.0, the same Asp and Glu residues are treated as deprotonated in correlation with a previous study on *P. tricornutum*[19]. For the m2 model, Glu86 is protonated, and the remaining Asp and Glu residues follow the same protonation pattern as in the Sm1 monomer. His84 was protonated at $N\epsilon$, while all other His residues at $N_\delta$ sites. In both Sm1 and m2 models, Chl-c 304 was treated protonated at lumenal pH 5.5 and deprotonated at lumenal pH 7, as it faces the lumenal side of the membrane with the acrylate group exposed to the acidic lumen. A lipid bilayer patch of approximately 350 thylakoid lipids[19], described by the AMBER force field[39], was used to embed each all-atom model. Lipid composition was based on the thylakoid membrane model of Chryasfoudi et al.[25] containing 45% MGDG, 25% DGDG, 25% SQDG, and 5% PG—reflecting an elevated SQDG/PG content (30%) relative to plant thylakoids (15–20%)[44]. The MGDG-DGDG lipid content is at 70% to simulate high-light adapted diatoms compared to low-light grown diatoms (50%)[44]. An amount of around 50000 TIP3P water molecules[45] was used for solvation, and each system included ~150 mM KCl with additional K$^+$ ions to neutralize protein and lipid charges. The equilibrated unit cell dimensions of each model were $16.3 \times 15.6 \times 8.5 \text{ nm}^3$ (monomer) as well as $18.8 \times 17.0 \times 8.8 \text{ nm}^3$ (tetramer).

### Setup of FCP-LHCX1 models

The crystal structure of the dimer fucoxanthin chlorophyll a/c -binding (FCP) protein from the diatom *P. tricornutum* (PDB ID: 6A2W) served as the starting template for building the FCP-LHCX1 dimer model. The LHCX1 protein 3D structure from *P. tricornutum* (uniprot code B7FYL0) was predicted by RosettaFold (robetta.bakerlab.org) using the default parametrization. The FCP-LHCX1 model was generated by structurally aligning the predicted LHCX1 onto the dimeric FCP scaffold resolved experimentally[14], by ChimeraX software and replacing the aligned FCP monomer by LHCX1. In the following we provide the LHCX1 sequence from the uniprot database.

>tr|B7FYL0 | B7FYL0_PHATC Protein fucoxanthin chlorophyll a/c protein OS=Phaeodactylum tricornutum (strain CCAP 1055/1) OX = 556484 GN=Lhcx1 PE = 3 SV = 1

**MKFAATILALIGSAAA**FAPAQTSRASTSLQYAKEDLVGAIPPVG FFDPLGFAD KADSPTLKRYREAELTHGRVAMLAVVGFLVGEAVEG SSFLFDASISGPAITHL SQVPAP FWVLLTIAIGASEQTRAVIGWVDPA DAPVDKPGLLRDDYVPGDLGF DPLGLKPSDPEELITLQTKELQNGR LAMLAAAGFMAQELVNGKGILENLQG

The targeting sequence of the Lhcx1 protein (marked in bold) was removed after the prediction for the subsequent MD modeling. The associated part of the polypeptide could have been found at the aquatic phase of the stomal side of the thylakoid membrane, facing away from the FCP complex (Fig. 4A–C) thus it should not interfere with the FCP-LHCX1 complex cross section. The resulting FCP-LHCX1 complex closely matches the model previously proposed in the literature[25]. All polypeptide chains and pigments were described using the same force fields as in the previous section. The Dtx (diatoxanthin) pigment was parametrized by modifiying the Ddx (diadinoxanthin) parameters specifically by removing one oxygen atom and adapting accordingly. The LHCX1 was considered non-pigmented[46]. Protonation states for the FCP and LHCX1 monomers were assigned as follows: all lumen-facing Asp and Glu residues were protonated to reflect protonation state at pH 5.5. In contrast, at pH 7.0, the same Asp and Glu residues were treated as deprotonated. For the LHCX1 monomer, Glu72, Glu189, and Asp79 were treated as protonated at low pH. All Histidine residues were protonated at $N_\delta$ sites.

### Classical molecular dynamics

Following established protocols, all systems were gradually relaxed and equilibrated by progressively releasing positional restraints on the heavy backbone atoms of the protein[19]. During a sequence of simulations in the NVT and NPT ensembles (constant volume and pressure, respectively), the system temperature was gradually raised from 100 to 303 K before entering the production phase. Classical molecular dynamics (MD) simulations were carried out using the leapfrog integrator available in GROMACS 2021[47] with a 2.0 fs integration time step. The production runs were performed in the constant pressure NPT ensemble with semi-isotropic couplings in the xy membrane plane and in the z-direction (compressibility at $4.5 \times 10^{-5}$ bar$^{-1}$). Furthermore, the van der Waals interactions were smoothly shifted to zero between 1.0 and 1.2 nm using the Verlet cutoff scheme. Short-range electrostatics were cut off at 1.2 nm, while long-range electrostatic interactions were computed using the particle mesh Ewald (PME) method[48,49]. All bonds between hydrogen atoms and heavy atoms were constrained using the LINCS algorithm[50]. The v-rescale thermostat was used[51] (303 K, temperature coupling constant 0.5) along with the C-rescale[44] for equilibration, while the Parrinello–Rahman barostats[52] was used for production (1 atm, pressure coupling constant 2.0). Independent trajectories (replicas) were initialized from structures extracted at 10 ns intervals during the final phase of equilibration. Simulation parameters were otherwise consistent with those applied in a prior study of *P. tricornutum*[19]. The total simulation time for the FCP models amounts to 12 µs, including four independent 0.5 µs trajectories for the tetramer model and four independent 0.5 µs trajectories for each monomeric model (Sm1 and m2), for two distinct pH protonation states (neutral – low). This simulation time can be translated in monomer-equivalent dynamics of: 2 pH states × (4 monomers × 2 µs + 1 monomer × 2 µs) = 20 µs sampling for the Sm1 monomer and 2 pH states × (1 monomer × 2 µs) = 4 µs sampling for the m2 FCP monomer in the different multimeric states. The simulations sum to 24 µs monomer-equivilant

dynamics for *C. gracilis*. The first 100 ns from each trajectory were considered as further equilibration, and the analysis was only performed for the final 400 ns of each independent trajectory. Structures were collected every 1.0 ns for all the trajectories. The total simulation time for the FCP-LHCX1 models amounted to 8 μs, including: four independent trajectories of 1 μs for the low pH and four independent trajectories of 1 μs for the neutral pH.

For details on the simulations for the isolated FCP from *P. tricornutum* (neutral-low pH) and the method of calculation of excitonic couplings between Chl-a 409/ Fx-301 refer to a previous study[19].

### Markov State modeling analysis (MSM) and dimensionality reduction

Dimers and tetramers were disassembled into different monomeric units before the analysis. The first 100–200 ns were disregarded from each individual trajectory based on the time when the backbone root-mean-square-deviation (RMSD) reaches a plateau, resulting cumulatively in 66.3 μs for two diatoms: *C. gracilis* and *P. tricornutum*. Only the FCP complex (common minimum core, without protons, ions or pigments) was extracted from the trajectories and all frames ($1\ ns^{-1}$) were structurally aligned based on Ca atoms by the GROMACS toolbox (trjconv -fit rot+trans) on a reference common core to assure consistency in the analysis. The PyEMMA package in Jupyter notebooks was employed[53]. All backbone torsional angles of residues aa 109–131 were chosen as input features. The dimensionality of the configurational space sampled in MD was further reduced. This was achieved by the time-lagged independent component analysis (tICA) to remove any redundant information. A 6-component tICA space and a time lag of 50 ns was used for coarse graining the degrees of freedom and identify a set of the slowest modes among all the initial input features (6 vectors). Then different MSMs were constructed with their slowest implied timescales to converge quickly and to be constant within a 95% confidence interval for lag times above 50 ns. The MSM passed the Chapman–Kolmogorov test at 95% confidence. The validation procedure is a standard approach in the field[23]. A lag time of 50 ns was thus selected for Bayesian MSM model construction. tICA components are the optimal linear combination of input features which maximizes their kinetic variance. The conformations of the FCPs (common core) were projected on the first two tICA vectors (IC-1, IC-2) and the trajectory frames were clustered into 100 cluster-centers (macrostates) by k-means clustering, as implemented in PyEMMA. The resulting macrostates were further coarse grained into a smaller number of four macrostates using PCCA++ as implemented in PyEMMA (conformations C1, C2, C3 and C4).

### Alternative dimensionality reduction to avg. angles and tilts vectors

a-helical segments of FCP were identified sequentially by the DSSP method (Define Secondary Structure of Proteins) implemented within the biopython toolbox (Bio.PDB.DSSP) and based on α-helical secondary structure annotations. For each a-helix, a principal component analysis (PCA) was applied to the Cα atom coordinates employing the scikit-learn toolbox to determine the dominant helical axis. Subsequently, the inter-helical angles ($\theta_1$, $\theta_2$, $\theta_3$, Fig. 1B) and tilt angles of the helices with respect to the z-axis (membrane normal) were computed.

### Random forest and gradient boost machine learning models

Due to the size of our data set for the experimentally resolved structures (118), the Random Forest approach is chosen for model training and subsequent predictions. The scikit-learn toolbox was employed for the analysis. The experimentally resolved set2 structures were clustered into four clusters by the k-means algorithm. A Random Forest (RF) machine learning model was trained on the experimentally resolved structures along with the four MSM-predicted conformations (C1-4) with avg. angle, avg. tilt as input features and the classification into the k-means clusters as output targets. This latter classification was performed at a higher level of accuracy,

employing further dimensionality reduction by Uniform Manifold Approximation and Projection (UMAP), compared to the simpler k-means on the un-reduced data. Thus, prior to training, feature dimensionality was further reduced using UMAP with Euclidean distance, 10 nearest neighbors, and a minimum distance of 0.2 to enhance classification performance. The resulting model was evaluated using a standard train-test split and classification metrics. The RF f1-scores were at 1.00 (red), 0.80 (orange), 0.73 (blue) and 0.92 (green) with an overall accuracy of 0.88. Without the use of the UMAP dimensionality reduction, the overall f1-scores are 1.00 (red), 0.83 (orange), 0.80 (blue) and 0.67 (green) with the overall accuracy at 0.84.

The Gradient Boost model was trained on 80% of data (input features/target excitonic couplings) and evaluated on the remaining 20%, yielding performance metrics including mean squared error (MSE) and the coefficient of determination ($R^2$).

### Reporting summary

Further information on research design is available in the Nature Portfolio Reporting Summary linked to this article.

### Data availability

All data generated or analyzed during this study are included in this published article. The numerical datasets used in this study (in ascii, delimited text format) and the FCP conformations (in pdb format) have been deposited in an open access github repository (https://github.com/vdas-upatras/fcp_diatoms).

### Code availability

All codes, along with code description (text files), used in the analyses are available as python scripts, or jupyter notebooks and have been deposited in an open access github repository (https://github.com/vdas-upatras/fcp_diatoms).

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

## Acknowledgements

The work and open access fees are funded by the Hellenic Foundation for Research & Innovation (H.F.R.I) in the context of the call "Basic Research Financing (Horizontal support for all Sciences), National Recovery and Resilience Plan (Greece 2.0) for the project number 014775, with acronym "SUNDIAL". B.Z. and S.M. have received funding from the European Union's Horizon Europe Research and Innovation Program under the Marie Skłodowska-Curie grant agreement No 101119442. Moreover, this work was also supported by computational time granted from the National Infrastructures for Research and Technology S.A. (GRNET S.A.) in the National HPC facility - ARIS - under project ID "FCPC". Part of the simulations were performed on a compute cluster funded through the DFG project INST 676/7-1 FUGG.

## Author contributions

Conceptualization: V.D.; Methodology: V.D. and U.K.; Investigation: T.-I.S., B.Z., and S.M.; Visualization: V.D., T.-I.S., and B.Z.; Supervision: V.D.; Writing-original draft: V.D.; Writing-review & editing: V.D. and U.K.; Molecular Dynamics & Machine Learning: I-T.S. and B.Z.; Calculation of Excitonic Couplings: S.M.

## Competing interests

The authors declare no competing interests.
