## [Transparent Peer Review file · Communications Chemistry]

Conformational Plasticity Enables Functional Switching in Diatom Light-Harvesting Complexes

Corresponding Author: Professor Vangelis Daskalakis

Version 0:

Reviewer comments:

Reviewer #1

(Remarks to the Author)

The current manuscript uses modeling approaches including machine learning to predict conformational states of diatom antennae. Given the relatively high amount of diatom LHC structures having been published in the last, roughly 5 years, there is now some data available to train machine learning algorithms. Such approaches are certainly interesting and add an additional layer of insights compared to the more classical biochemical, physiological and molecular biology approaches.

I am not an expert at all in this kind of approach. And that is already my major issue with this article. It addresses questions definitely relevant for understanding the biochemistry and physiology of diatoms and here particularly the regulation of photosynthesis. However, it is written from the perspective of a physicist (or mathematician), which is normal as this kind of approach requires the respective skills. However, the paper will hardly be of interest for other modelers (who could easily understand all the different parameters), but the target group is a largely different one. But then the manuscript somehow needs to be better transferred to the level of a diatom physiologist/biochemist, so that we are able to understand better the relevant output but also the methodology behind. I hence strongly advise that the authors undertake significant efforts to make their approach and especially their results more understandable, e.g. with 1 or 2 introducing sentences for each result.

Two examples for the first result of the manuscript:

L122: "The reweighted free energy surface (FES) was projected onto the slowest degrees of freedom of the FCP common core, determined by the time-lagged independent component analysis (tICA) procedure and described by the two vectors IC-1 and IC-2. This is shown in Fig. 1A, alongside the positions of the four identified kinetically distinct states 1-4."

What are these two vectors IC-1 and IC-2?

What is a time-lagged independent component analysis and why it is applied?

On the same page, Figure caption Fig. 1: "The avg. angle θ is defined as the average of the angles θ_1 , θ_2 and θ_3 defined by the vectors: v1 aa 64-104, v2 109-116, v3 117-131 and v4 for aa 149-174. The avg. tilt is calculated as the average of the angles formed by each vector v1 to v4 with respect to the membrane normal Z. "

What are these angles describing in a more general way?

As I cannot evaluate the modelling process itself, in the following I will only refer to the biochemistry and physiology of diatom antenna and the general style of the manuscript, where I have a few points to consider.

- In general, I would strongly suggest that the authors also try to model their system with a pigmented Lhcx (if timewise currently not possible, then please in the future). They assume that Lhcx does not contain pigments based on the study by Giovagnetti et al (2022). However, the highly related Lhcsr of green algae bind pigments (e.g. Bonente et al. 2011, doi:10.1371/journal.pbio.1000577), Lhcx of diatoms contain most motifs found in the other Lhcs that bind pigments, and the 1:1 interdependency of diatoxanthin and NPQ also strongly argues for a pigment binding. To my mind it is highly unlikely that Lhcxs do not bind pigments. As there is yet no final experimental evidence for either of the two possibilities (pigment binding or not), the modelling should also address both options.

- I am not sure whether the journal requirements are different here, but usually a species is written in italics, the genus name is only abbreviated with the first letter, and the species specification is in minuscules.

- I do not know whether it eventually affects the modelling, but it would have been wise to remove the protein targeting sequence of the Lhcx1 protein before starting the modelling. This helix is not present once the Lhcx proteins are at their final destinations in the thylakoids.

- L191: "In order to answer this question, we focus on a special pigment pair in the FCP structure of *Ph. tricornutum* (Chl-193 a and carotenoid Fx-301) that has been proposed computationally and experimentally to be involved in such a transition 19,24,25. Under NPQ conditions, energy transfer from Chl-a to Fx enables a quick dissipation of the excess absorbed energy as heat, through a short-lived excited state of Fx. The excitonic coupling value between Chl-a (m) and Fx (n) is a measure of the efficiency of this energy transfer, with the rate of transfer given by $k_{mn} \approx |V_{mn}|^2$ 19, where V_{mn} is the excitonic coupling between m and n."

Although this is certainly interesting, the authors should be more careful in assigning this interaction as essential for NPQ. The cited references are mainly their own (computational) works. The only experimental study cited (Agostini et al. 2023) never employs the term NPQ for this Fx-Chl a pair. They rather assign this as an important triplet Chl decay channel, another function carotenoids are famous for. I personally doubt that this Chl-Fx channel is the one responsible for NPQ in diatoms. The role of Dtx in this process is simply too essential as to transfer it to Fx.

- In the discussion I would like to have a better interpretation of these results in light of the existing experimental results. I find it kind of unusual that in the discussion there is not any reference to any other work.

Things to be discussed:

What do these simulations add on top of the known experimental results?

In which view do they expand our understanding of the light harvesting and/or photoprotection process in diatoms?

Is there anything the modeling predicts they may be tested by a special experimental wet bench approach in the future?

Reviewer #2

(Remarks to the Author)

The paper 'Conformational Plasticity Enables Functional Switching in Diatom Light-Harvesting Complexes' by Daskalakis et al reports an interesting molecular dynamic study about FCP from diatoms. Altogether, the paper is well-written, the methodology partly inventive, and the results are appealing. I have only few rather positive comments on this paper, and I suggest it to be published after minor revision.

Minor comments :

1. The authors use monomer-centric analysis. They reduce (trimers and tetramers from the PDB) FCPs to monomer cores for MSM/ML. While this certainly results in a significant reduction of the computation time, this ignores interface constraints and inter-monomer excitonic cross-talk; the C1–C4 states may not persist in intact trimers/tetramers. The authors should make it clear whether this induces limitations in the landscape visited by the FCP and which ones.

2. In a way, they equate modest increases in a single Chl–Fx coupling to energy transfer, hence to NPQ. Higher coupling alone does not ensure quenching, and additionally the reported changes are from 7 to 30 cm⁻¹, which are not a huge difference. Moreover, trapping requires the energy to be transferred from Chl to the carotenoid, thus involving a change in the energetics of latter

3. Although we have little molecular, experimental, information on the FCP in their quenching state, last year Alexandre et al reported in BBA Bioenergetics that the aggregation of FCP induces their quenching and is accompanied by small structural changes in some of the bound carotenoids. Do the author see corresponding changes in their QMD experiments ?

Reviewer #3

(Remarks to the Author)

I co-reviewed this manuscript with one of the reviewers who provided the listed reports. This is part of the Communications Chemistry initiative to facilitate training in peer review and to provide appropriate recognition for Early Career Researchers who co-review manuscripts.

Version 1:

Reviewer comments:

Reviewer #1

(Remarks to the Author)

The authors addressed my concerns absolutely satisfactory. It is now well readable also for an outsider of MD and I am happy to now be able to extract the interesting and relevant information for my field of research.

Reviewer #2

(Remarks to the Author)

The authors mostly completed and modified their manuscript according to the instructions of this reviewer. The paper is now in an acceptable form. I recommend it to be accepted without further modification.

Reviewer #3

(Remarks to the Author)

I co-reviewed this manuscript with one of the reviewers who provided the listed reports. This is part of the Communications Chemistry initiative to facilitate training in peer review and to provide appropriate recognition for Early Career Researchers who co-review manuscripts.

From: Vangelis Daskalakis/ Sept 28, 2025
Reply to reviewers/ Ref: COMMSCHEM-25-0739

Dear Editor,

Thank you for your letter of Sept 17th, 2025, with referee's reports and the opportunity to submit a revised version of our manuscript. We are pleased that both referees find our manuscript suitable for Communications Chemistry journal. We are also thankful for the criticism and suggestions by both reviewers that helped us in preparing an improved version of our manuscript. In the following, we provide a point-to-point answer to all queries along with a revised version of the manuscript with changes highlighted in yellow.

Reviewer #1 (Remarks to the Author):

The current manuscript uses modeling approaches including machine learning to predict conformational states of diatom antennae. Given the relatively high amount of diatom LHC structures having been published in the last, roughly 5 years, there is now some data available to train machine learning algorithms. Such approaches are certainly interesting and add an additional layer of insights compared to the more classical biochemical, physiological and molecular biology approaches.

We thank Reviewer #2 for the positive evaluation and for her/his appreciation of our work.

I am not an expert at all in this kind of approach. And that is already my major issue with this article. It addresses questions definitely relevant for understanding the biochemistry and physiology of diatoms and here particularly the regulation of photosynthesis. However, it is written from the perspective of a physicist (or mathematician), which is normal as this kind of approach requires the respective skills. However, the paper will hardly be of interest for other modelers (who could easily understand all the different parameters), but the target group is a largely different one. But then the manuscript somehow needs to be better transferred to the level of a diatom physiologist/biochemist, so that we are able to understand better the relevant output but also the methodology behind. I hence strongly advise that the authors undertake significant efforts to make their approach and especially their results more understandable, e.g. with 1 or 2 introducing sentences for each result.

Two examples for the first result of the manuscript:

L122: "The reweighted free energy surface (FES) was projected onto the slowest degrees of freedom of the FCP common core, determined by the time-lagged independent component analysis (tICA) procedure and described by the two vectors IC-1 and IC-2. This is shown in Fig. 1A, alongside the positions of the four identified kinetically distinct states 1-4."

What are these two vectors IC-1 and IC-2?

What is a time-lagged independent component analysis and why it is applied?

On the same page, Figure caption Fig. 1: "The avg. angle θ is defined as the average of the angles θ_1 , θ_2 and θ_3 defined by the vectors: v_1 aa 64-104, v_2 109-116, v_3 117-131 and v_4 for aa 149-

174. The avg. tilt is calculated as the average of the angles formed by each vector v1 to v4 with respect to the membrane normal Z. “

What are these angles describing in a more general way?

We thank Reviewer #1 for his/her insightful observation that our terminology might be too technical to the non-experts on computational biophysics or bioinformatics. Therefore, we have revised the results section, to include a more understandable version of our approaches. We have added the following to the revised manuscript:

*lines 123-139/ A Markov state model (MSM), a statistical analysis tool, was constructed from the MD trajectories, which can predict the long-time scale behavior of a biomolecular system based on short MD simulations²³. In order to describe the main conformational changes of the FCP complex over data set 1, one can rely on the complete array of backbone torsional angles. However, this huge array contains a lot of noise, i.e. thermal fluctuations and fast motions. Therefore, the time-lagged independent component analysis (tICA) was used to reduce the dimensionality of this array, a method that extracts the slowest motions from the simulations. These motions, captured by two vectors (IC-1 and IC-2), represent the most biologically relevant rearrangements of the protein-pigment complex. IC-1 and 2 consist of linear transformations of the high-dimensionality space to a subspace that maximally preserves the kinetic content. In our case, IC-1 and 2 represent linear combinations of torsional angles on the FCP backbones. For further details refer to the Methods. FES was projected along these vectors of the FCP common core to identify the most stable conformations. This is shown in **Fig. 1A**, alongside the positions of the four identified kinetically distinct states 1-4, or ensembles of conformations. State-2 is shared between the FCPs of *P. tricornutum* and *C. gracilis*, states-1 and- 3 are assigned to *P. tricornutum* whereas state-4 to *C. gracilis*. Four conformations, as ensemble averages per state, termed macrostates, that are associated with states 1-4 were also predicted by MSM (C1 to C4) [...]*

*lines 154-167/ The avg. angle $\bar{\theta}$ describes the relative orientations of key transmembrane helices within the FCP complex, providing a measure of their concerted bending or splaying. The avg. tilt quantifies how these helices are oriented relative to the membrane plane, thereby reflecting potential conformational shifts that could modulate pigment-pigment interactions relevant for light harvesting and photoprotection. Moreover, the avg. angle $\bar{\theta}$ and avg. tilt might also capture transitions between partial helicity or random coil structures, especially for the luminal side of the FCP complex (vector v_2) as these transitions would shift the avg. values. Together, these parameters summarize protein conformational changes, like scaffold expansion (or swelling), that can influence pigment interactions. The probability distribution $P(x,y)$ of the MD data set1 over the two reduced dimensions is converted into a free energy landscape by $-\ln(P(x,y))$ (energy in units of kT) (**Fig. 1C**). This representation also highlights the most stable states of the protein, which correspond to the minima on the surface. The landscape in **Fig. 1C** shows roughly the same conformational space as in **Fig. 1A** but arranged in a rotated orientation [...]*

*lines 187-202/ For further analysis, we have used machine learning approaches, namely algorithms that can detect patterns in complex data like in an elaborate fitting process, without needing an explicit equation. Supervised models rely on a well-established set of input features (like the avg. angle and avg. tilt), whereas unsupervised models do need the definition of specific features to identify patterns. By training these models on our simulation results (input features), we can uncover relationships (output) between protein structure and pigment interactions that would be difficult to capture with simple analytical approaches. For details refer to the Methods section. The experimentally resolved set2 structures (**Fig. 1E**) were classified (grouped) into four clusters identified by color (**Fig. 1F**) and based on their structural diversity. This was done by unsupervised classification which was performed by a simple k-means clustering over the avg. angle and avg. tilt features. K-means clustering is a way of grouping similar protein conformations together based on their structural features. The algorithm looks for patterns in the data and assigns each conformation to one of several clusters, such that conformations within the same cluster are more similar to each other than to those in other clusters. In practice, this means that instead of analyzing millions of individual simulation frames, we can summarize the dynamics into a handful of representative structural ‘states,’ which correspond to the cluster centers [...]*

lines 207-212/ RF approach is like asking many independent non-linear fitting algorithms ('decision trees') the same question (e.g. the value of a y(x) dependent variable) and then averaging their answers for the output. Each tree looks at the data, or fits the data in a slightly different way, and together they provide a reliable consensus. This helps reduce the chance of over-fitting to noise and highlights which structural features most strongly affect FCP conformations [...]

lines 246-259/ Gradient boosting regression also uses decision trees, but in this case each tree is built step by step, learning from the mistakes of the previous one. This gradual refinement allows the model to capture subtle relationships in the data, making it powerful for predicting how small structural changes influence inter-pigment coupling.

As I cannot evaluate the modelling process itself, in the following I will only refer to the biochemistry and physiology of diatom antenna and the general style of the manuscript, where I have a few points to consider.

- In general, I would strongly suggest that the authors also try to model their system with a pigmented Lhc (if timewise currently not possible, then please in the future). They assume that Lhc does not contain pigments based on the study by Giovagnetti et al (2022). However, the highly related Lhcsr of green algae bind pigments (e.g. Bonente et al. 2011, doi:10.1371/journal.pbio.1000577), Lhc of diatoms contain most motifs found in the other Lhcs that bind pigments, and the 1:1 interdependency of diatoxanthin and NPQ also strongly argues for a pigment binding. To my mind it is highly unlikely that Lhcs do not bind pigments. As there is yet no final experimental evidence for either of the two possibilities (pigment binding or not), the modelling should also address both options.

Indeed, there is no definitive experimental proof that LHCX1 is either pigmented, or not. While there exists no definitive answer to the question, we plan to develop a model of a pigmented LHCX1 in future and ongoing work. Nevertheless, this modeling work requires long simulation times and cross-validation via spectroscopic studies. Thus, we prefer leaving it for future work. The revised manuscript now reads:

Lines 313-318/ We have modelled an LHCX1 structure without pigments. Modeling a fully pigmented LHCX1 would require a separate study, since the pigment composition and precise pigment binding locations, if any, are not yet well established in the literature. Such an effort would involve building multiple structural models with different pigment arrangements, supported by biochemical assays and spectroscopic data from isolated LHCX1. We plan to address this pigmented model in a future work.

- I am not sure whether the journal requirements are different here, but usually a species is written in italics, the genus name is only abbreviated with the first letter, and the species specification is in minuscules.

Thank you for pointing this out. We have changed the abbreviations as requested, throughout the revised manuscript and selected panels of Fig. 1.

- I do not know whether it eventually affects the modelling, but it would have been wise to

remove the protein targeting sequence of the Lhcx1 protein before starting the modelling. This helix is not present once the Lhcx proteins are at their final destinations in the thylakoids.

The observation is logical, indeed. In the original version, the methods section contains the complete sequence of LHCX1 in *P. tricornutum* as provided by the UNIPROT database. However, we failed to mention that the targeting sequence was taken into consideration only for the structural prediction by AlphaFold, but it was removed before the model setup and MD simulations. The revised manuscript is updated accordingly in the methods section:

Lines 485-497/ In the following we provide the LHCX1 sequence from the uniprot database.

```
>tr|B7FYLO|B7FYLO_PHATC Protein fucoxanthin chlorophyll a/c protein OS=Phaeodactylum tricornutum (strain CCAP 1055/1) OX=556484 GN=Lhcx1 PE=3 SV=1
MKFAATILALIGSAAAFAPAQTSRASTSLQYAKEDLVGAIPPVGFFDPLGFADKADSPTLKRYREAELTHGRVAMLA VVGFLVGEAVEGSS
FLFDASISGPAITHL SQVPAP FWVLLTIAIGASEQTRAVIGWVDPADAPVDKPGLLRDDYVPGDLGFDPLGLKPSDPEELITLQTKELQN
GRLAMLAAAGFMAQELVNGKGILENLQG
```

The targeting sequence of the Lhcx1 protein (marked in bold) was removed after the prediction for the subsequent MD modeling. The associated part of the polypeptide could have been found at the aquatic phase of the stomal side of the thylakoid membrane, facing away from the FCP complex (Fig. 4A-C) thus it should not interfere with the FCP-LHCX1 complex cross section.

- L191: “In order to answer this question, we focus on a special pigment pair in the FCP structure of *Ph. tricornutum* (Chl-193 a and carotenoid Fx-301) that has been proposed computationally and experimentally to be involved in such a transition 19,24,25. Under NPQ conditions, energy transfer from Chl-a to Fx enables a quick dissipation of the excess absorbed energy as heat, through a short-lived excited state of Fx. The excitonic coupling value between Chl-a (m) and Fx (n) is a measure of the efficiency of this energy transfer, with the rate of transfer given by $k_{mn} \approx |V_{mn}|^2 / 19$, where V_{mn} is the excitonic coupling between m and n.”

Although this is certainly interesting, the authors should be more careful in assigning this interaction as essential for NPQ. The cited references are mainly their own (computational) works. The only experimental study cited (Agostini et al. 2023) never employs the term NPQ for this Fx-Chl a pair. They rather assign this as an important triplet Chl decay channel, another function carotenoids are famous for. I personally doubt that this Chl-Fx channel is the one responsible for NPQ in diatoms. The role of Dtx in this process is simply too essential as to transfer it to Fx.

We thank Reviewer #1 for prompting us to elucidate further this aspect of the work. Identifying the site of quenching experimentally is not always straight-forward. Thus far, there is no experimental work to definitively point to the site of quenching within the PSII-FCPII super complex (i.e a single pigment pair, or special interaction). Therefore, we have revised our results and discussion section to elaborate on our choice of the Chl-a 409/ Fx-301 pigment pair and provide further evidence that our model takes Dtx also into consideration for NPQ in diatoms (see our reply to the comment #3 by Reviewer #2 further below). The revised manuscript now reads:

Lines 76-78/ This integrated computational approach allows for a quantitative assessment of how FCP conformation changes in response to external stimuli which is likely associated with different acclimation states [...]

*Lines 223-233/ In the absence of strong experimental evidence on the actual NPQ site (specific pigments) within FCPs,²⁴ we can rely only on theoretical predictions and an experimental study^{19,25,26}. We focus on a pigment pair in the FCP structure of *P. tricornutum* (Chl-a 409 and carotenoid Fx-301) that has been proposed computationally as the only pigment pair to exert pH-dependent fluctuating excitonic coupling values between 7-40 cm⁻¹ and experimentally to be involved in an important triplet Chl decay channel^{19,25,26}. Under NPQ conditions, energy transfer between Chl-a and carotenoids (Cars) like Fx or Dtx could enable a quick dissipation of the excess absorbed energy as heat, through a short-lived excited state of Car, in an analogy to higher plants where the major LHC of Photosystem II binds the Chl-a 612/ Lutein pigment pair and energy can be dissipated by energy transfer from Chl-a to Lutein^{27,28} [...]*

Lines 237-243/ Low V_{mn} values (<10 cm⁻¹) indicate rather weak Chl-Car interactions and higher values (>10 cm⁻¹) are characteristic for efficient energy transfer, that might relate with a component of the NPQ dissipating mechanism¹⁹. In fact, aggregation of FCPs significantly enhanced excitation-energy quenching by changes also in Chl-Car interactions that affected energy transfer dynamics, as probed by absolute fluorescence spectroscopy and fluorescence decay-associated spectra.²⁹ The authors of the latter study also suggest that Chls may interact with nearby fucoxanthins, resulting in excitation quenching [...]

*Lines 258-262/ The coupling values roughly follow the progression of the FCP scaffold expansion or swelling as depicted in **Fig. 1B** (C1-C4 to C2 and to C3). We find that certain conformations slightly enhance the interaction between Chl-a 409 and Fx-301 pigments, by an increase in Chl-Fx excitonic coupling values. While this alone does not prove that quenching occurs for C3, it points to a structural mechanism that likely enhances energy dissipation under photoprotective conditions.*

- In the discussion I would like to have a better interpretation of these results in light of the existing experimental results. I find it kind of unusual that in the discussion there is not any reference to any other work.

Thank you for pointing this out. We have now added more references, and we discuss our results in view of the associated recent studies that are relevant to our work (**24. BBA - Bioenergetics** 2024, 1865, 149500; **29. J Phys Chem B** 2025, doi:10.1021/acs.jpcc.4c06894; **38. PNAS** 2017, 114, E11063–E11071). We also provide the new Fig. 4D which describes the dynamics of the Dtx pigment in the FCP-LHCX1 complex that compare well with the experimental literature. In the following we address each question raised by Reviewer #2 by providing parts of the revised manuscript.

Things to be discussed:

What do these simulations add on top of the known experimental results?

In which view do they expand our understanding of the light harvesting and/or photoprotection process in diatoms?

*Lines 338-344/ Recent experimental evidence suggests that Dtx displays significant conformational changes for high-light treated FCPa upon aggregation.²⁴ Experimental studies in the literature^{6,34-37} also indicate that LHCX1 and Dtx are necessary components for the induction of NPQ in diatoms. In summary, our results show that these components also fine tune the dynamics of the FCP scaffold and pigment interactions therein. In the case of a pigmented LHCX1, the different Dtx orientations (**Fig. 4C-D**) could activate energy dissipating pathways between Chls bound in LHCX1 and Dtx [...]*

Lines 374-379/ Our simulations provide a molecular-level complementary to experimental observations of the FCP plasticity. In line with the experiential literature,³⁸ we have identified that the light harvesting antennas in diatoms (FCP scaffold) are quite flexible. While time-resolved spectroscopy and cryo-EM have revealed static structures and ensemble energy transfer dynamics, our approach captures the conformational fluctuations and inter-pigment exciton couplings that could underlie such processes [...]

*Lines 394-400/ Our study aligns well with a recent experimental work on FCP aggregates that show fluorescence quenching.²⁴ In this latter study, under NPQ conditions, Dtx exerts considerably stronger conformational changes compared to Ddx in line with **Figs. 4C-D** herein. We go beyond this observation to identify the mechanism behind these changes as being the FCP-LHCX1 interaction or possible FCP-FCP aggregation with the same interface as the FCP-LHCX1 complex²⁵ (see also the methods section) [...]*

Lines 404-409/ The simulations suggest that small shifts in the relative position (angle, tilt) of FCP helices can change how strongly chlorophylls and fucoxanthins interact. This provides a possible structural link between protein plasticity and the balance between light harvesting and photoprotection. We emphasize that our results highlight a possible conformational change that is associated with increased coupling between chlorophylls and fucoxanthins. We note that experimental evidence has also suggested that Fx carotenoids contribute to quenching.^{24,29} [...]

Is there anything the modeling predicts they may be tested by a special experimental wet bench approach in the future?

Lines 416-428/ Our predictions could be experimentally tested using site-directed mutagenesis that can bias helix packing by introducing either Cys pairs for cross-linking so that the FCP complex can assume expanded (swollen) scaffold, or by replacing bulky residues to Ala to favor a contracted scaffold. Measurements like Transient Absorption (TA) and Two-Dimensional Electronic spectra (2DES) can identify Chl-Fx exciton energy transfers therein. Furthermore, by collecting FCP/LHC sequences from public genomes sequence alignments to build phylogenies and associate such sequences with AlphaFold structural predictions. This would provide helix-packing metrics and applying phylogenetically aware statistics / machine learning one can associate sequence signatures with expanded or contracted scaffolds and diatom phenotypes (efficient light harvesters or with robust photoprotection). For example, species found under sea ice (polar) are adapted to even lower, blue-shifted light, while open-water species are better equipped to cope with light fluctuations and rapid changes, demonstrating diverse phenotypes among diatoms.

Reviewer #2 (Remarks to the Author):

The paper 'Conformational Plasticity Enables Functional Switching in Diatom Light-Harvesting Complexes' by Daskalakis et al reports an interesting molecular dynamic study about FCP from diatoms. Altogether, the paper is well-written, the methodology partly inventive, and the results are appealing. I have only few rather positive comments on this paper, and I suggest it to be published after minor revision.

We thank Reviewer #2 for the positive evaluation and for her/his appreciation of our work.

Minor comments :

1. The authors use monomer-centric analysis. They reduce (trimers and tetramers from the PDB) FCPs to monomer cores for MSM/ML. While this certainly results in a significant reduction of the computation time, this ignores interface constraints and inter-monomer excitonic cross-talk; the

C1–C4 states may not persist in intact trimers/tetramers. The authors should make it clear whether this induces limitations in the landscape visited by the FCP and which ones.

Thank you for the opportunity to provide further details on our reasoning behind this monomer-centric analysis. We would like to express our regret that this crucial point was not made clear in the original manuscript. Nevertheless, we have to note that the Molecular Dynamics simulations explicitly included the oligomerization state of the FCP models. The revised manuscript now reads:

Lines 86-91/ Our simulations included the complete dimeric, trimeric, or tetrameric FCP complexes within a model thylakoid membrane. However, only for the analysis, we focused on the behavior of individual monomers, being part of oligomers or not. All these together, provided a total of 80 μ s of monomer-level dynamics for FCPs from the two species. This ensures that the effects of pH and oligomerization are included in the results [...]

Lines 382-391/ Experimentally resolved structures of FCPs consist mainly of multimers (Fig.1D, E, F). In fact, although more balanced, also considerably more data from multimers are collected compared to monomers for the MD-based dynamics shown in Fig. 1C; for the same sampling of 2 μ s, we collect 2 μ s dynamics for the monomer, but 8 μ s monomer-equivalent dynamics for a tetramer. Thus, the Free Energy Surfaces shown in Fig 1 are more biased towards multimers. However, the FES shown in Fig. 4B even though it refers strictly to monomer only dynamics, yet it shows the same picture as the ensembles biased more towards the multimers. Furthermore, in our previous study¹⁹, we have shown that FCP monomers and multimers can share the same states. Cumulatively, these results point to a flexible FCP scaffold that can be fine-tuned by pH, LHCX1-Dtx and multimerization states, with this tuning to be leaving its signature on the experimentally resolved structures.

2. In a way, they equate modest increases in a single Chl–Fx coupling to energy transfer, hence to NPQ. Higher coupling alone does not ensure quenching, and additionally the reported changes are from 7 to 30 cm^{-1} , which are not a huge difference. Moreover, trapping requires the energy to be transferred from Chl to the carotenoid, thus involving a change in the energetics of latter

Thank you for pointing this out. We have revised the manuscript to include further details. The revised manuscript now reads:

Lines 404-416/ The simulations suggest that small shifts in the relative position (angle, tilt) of FCP helices can change how strongly chlorophylls and fucoxanthins interact. This provides a possible structural link between protein plasticity and the balance between light harvesting and photoprotection. We emphasize that our results highlight a possible conformational change that is associated with increased coupling between chlorophylls and fucoxanthins. We note that experimental evidence has also suggested that Fx carotenoids contribute to quenching.^{24,29} We observe a modest increase in Chl–Fx coupling (from 7.65 to 26.41 cm^{-1}), which however translates into a $\left(\frac{7.65}{26.41}\right)^2 \approx 12$ fold increase of the exciton transfer rate and also suggests a structural predisposition toward enhanced interaction between these pigments. However, stronger coupling alone does not ensure NPQ. Thus, our simulations should be interpreted as identifying a structural mechanism that could contribute to quenching, rather than defining it. Future combined computational-experimental work will be needed to assess whether the observed conformational states also shift the energetic landscape of Chl and Fx/ Dtx in a way that enables efficient energy dissipation.

3. Although we have little molecular, experimental, information on the FCP in their quenching state, last year Alexandre et al reported in BBA Bioenergetics that the aggregation of FCP induces their quenching and is accompanied by small structural changes in some of the bound carotenoids. Do the author see corresponding changes in their QMD experiments ?

We are grateful to Reviewer #2 for providing this key study that we regretfully overlooked when writing our manuscript. We have now included new data in the revised Fig. 4 (panels C-D) by analyzing further our MD trajectories. We observe a conformational change of the bound Dtx associated with the FCP-LHCX1 complex formation. The revised manuscript, along with Fig. 4 panels C-D now read:

Lines 353-360/

Figure 4 | [...] (C) Orientations of the Ddx (orange) and Dtx (red) carotenoids over the dynamics in the absence of LHCX1 interaction (Ddx) and the presence of LHCX1 interaction (Dtx). Chl-a 405 in the vicinity of Ddx/ Dtx is also shown for reference, (D) Distributions of the root mean square deviations (RMSD) of the carbon atoms of Ddx (orange) or Dtx (red) as violin plots for the different models. The inset shows the Root Mean Square Fluctuations (RMSF) of the Dtx/ Ddx carbon atoms for $\Delta(\text{LHCX1}, \text{Dtx})$. The atom numbering is shown for selected regions on two representative structures: Ddx (orange)-Dtx (red); the ring close to the stroma #1-9, the ring close to the lumen #32-40 and branching carbons.

Lines 322-340/ A comparison of **Fig. 4B** and **Fig. 2B** shows that the LHCX1 protein and the xanthophyll cycle activates the strong interaction between Chl-a 409 and Fx-301 by shifting the FCP population to expanded (swollen) FCP scaffolds and increased excitonic coupling values for the Chl-a 409/ Fx-301 pigment pair (cm^{-1}) in the diatoms *P. tricornutum* and *C. gracilis*. Furthermore, we observe that upon the FCP-LHCX1 interaction at low pH, the Dtx assumes various orientations (red structures in **Fig. 4C**) that are associated with the C3 conformation as shown in **Fig. 4C**. The Dtx orientation in the presence of LHCX1 is significantly more variable as also shown from the distribution of RMSD values as violin plots in **Fig. 4D** with respect to the FCP Ca atoms used as reference. Ddx orientation shows less variance between isolated FCPs at low and neutral pH (orange structures in **Fig. 4C** and **Fig. 4D**). Here, Dtx is modelled at the interface between LHCX1 and FCP (**Figs. 4A, C**) based on the original location of Ddx in the experimentally resolved crystal structure of the FCP from *P. tricornutum*¹⁴. Therefore, Dtx interactions with adjacent pigments can be tuned by the interaction with LHCX1. Chl-a 405 is found adjacent to Ddx/ Dtx thus LHCX1 can tune also the energy transfer dynamics between Chl-a 405 and Dtx. Moreover, the RMSF values for the carbon atoms of Ddx/ Dtx exert significant mobility in the transition from low pH (Ddx) to low pH (LHCX1+Dtx) indicated by the $\Delta(\text{LHCX1}, \text{Dtx})$ plot at low pH in the inset of **Fig. 4D**, along with the superposition of the main Ddx/ Dtx conformations along the dynamics sampled. Recent experimental evidence suggests that Dtx displays significant conformational changes for high-light treated FCPa upon aggregation.²⁴

An obvious further exploration would be to calculate excitonic coupling values between Ddx/ Dtx and Chl-a 405 (**Fig. 4C**). However, such calculations necessitate the calculation of partial charges

in Dtx/ Ddx at the transition state. Unlike the case of Fx, these charges are not readily available from our groups. Thus, calculations at a computationally demanding level of theory like the multireference configuration interaction (MRCI) method are needed for both Ddx and Dtx. Then we should apply the TrESP approach (*Phys. Chem. Chem. Phys.*, 2022, **24**, 5014-5038) to calculate excitonic coupling values between Ddx/ Dtx and Chl-a 405. This would take significant amount of time and furthermore, Ddx/ Dtx is not resolved experimentally in all structures considered in this work (i.e. FCP from *C. gracilis*) which would complicate our discussion on a conserved effect on excitonic couplings. Thus, we prefer not to include or discuss such an approach within the current manuscript.

Reviewer #3 (Remarks to the Author):

I co-reviewed this manuscript with one of the reviewers who provided the listed reports. This is part of the Communications Chemistry initiative to facilitate training in peer review and to provide appropriate recognition for Early Career Researchers who co-review manuscripts.

We are happy that our manuscript helped the training in peer review for Early Career Researchers.